# Factors Associated with Regular Zumba Practice as Preliminary Results: A Population-Based Approach in Cebu Province, the Philippines

**DOI:** 10.3390/ijerph18105339

**Published:** 2021-05-17

**Authors:** Junko Yamasaki, Kayako Sakisaka, Parolita A. Mission, Nasudi G. Soluta, Norre Jean V. delos Santos, Julie Vannie Palaca, Retz Pol O. Pacalioga

**Affiliations:** 1Graduate School of Public Health, Teikyo University, Tokyo 173-8605, Japan; solxjunko@yahoo.co.jp; 2National Nutrition Council Region VII, Cebu City 6000, Philippines; letlet.mission@nnc.gov.ph (P.A.M.); nasudi.soluta@nnc.gov.ph (N.G.S.); norrejean.delossantos@nnc.gov.ph (N.J.V.d.S.); vpalaca@gmail.com (J.V.P.); retzpolpacalioga@gmail.com (R.P.O.P.)

**Keywords:** Philippines, Cebu city, physical activity, Zumba, exercise, non-communicable diseases

## Abstract

The prevalence of overweight/obesity in the adult population in the Philippines has doubled in the past 20 years. Zumba exercise has recently been implemented throughout the Philippines. However, there is scarce information on the effects of Zumba on obesity and Zumba participants’ characteristics in the Philippines. This study described the current practice of Zumba in the Philippines, along with the practitioners’ characteristics, and identified factors associated with Zumba participation. In this observational, cross-sectional study, a structured questionnaire was used to survey 10 Zumba locations in September 2019. Anthropometric measurements of participants were assessed. Respondents included 171 women (88.6%) and 22 men (11.4%), with a mean (±standard deviation [SD]) age of 44.1 (±8.9) years. All respondents answered that Zumba was enjoyable, and some answered “very enjoyable”. Determinants of frequent participation were as follows: being older than the mean age of participants, starting Zumba to enjoy dancing, starting Zumba not to lose weight, shopping mall location, and participation fee required. “To enjoy dance” being a motivation for Zumba practice was identified as a determinant of frequent participation rather than “to lose weight.” The element of “enjoyable” may strongly influence the continuation and frequent participation of Zumba exercise in the Philippines.

## 1. Introduction

The increase in the prevalence of non-communicable diseases (NCDs) is a major global concern. According to the World Health Organization (WHO) Fact Sheet published in 2018 [1], approximately 71% of deaths globally (among approximately 41 million people) were associated with NCDs, and approximately 75% of those deaths occurred in residents of low/middle-income countries [2,3]. The basic strategy against NCDs consists of two pillars: prevention and control. The WHO aims to reduce the incidence of NCD-related deaths among the world’s population aged 30–70 years by 25% before the year 2025 [2,4].

In the Republic of the Philippines (hereafter referred to as the Philippines), there has been a recent drastic transition in the disease structure. In addition to the deaths caused by conventional “infections,” the incidence of deaths due to lifestyle diseases has shown an increasing trend, and this is related to a mixed situation of health problems in developing countries as well as in developed countries (i.e., the dual structure of diseases) [5,6]. The incidence of overweight/obesity in the adult population in the Philippines has doubled in the past 20 years, and approximately one-fifth of the adult population was reported to have high blood pressure (20.4% among males and 17.7% among females, WHO, Geneva, Switzerland, 2015) [7]. The disease structure change in the Philippines is closely related to the change in diet or lack of exercise due to the effects of globalization on the nation [8,9].

In response to this situation, the Philippines government has recognized the significance of the cross-department approach for addressing the issue at national and municipal levels and implemented various initiatives to enhance the national healthcare system [10]. Zumba exercise (hereafter referred to as Zumba) has recently been implemented throughout the Philippines since around 2014 as a government-initiated population approach measure for lifestyle disease prevention. Zumba is offered at various regional-level health promotion events in public spaces such as shopping malls, parks, hotels, and barangays (the smallest administrative division in the Philippines) for free or for a fee of 20 pesos (equivalent to approximately 0.4 USD) [10].

Zumba is a Latin exercise that originated in Latin America, which has rapidly increased in popularity since around 2001, and there are approximately 14 million people across 150 countries participating in Zumba programs currently [11]. The concept of Zumba is to “enjoy music and move the body freely”, targeting a wide range of age groups, which is sufficiently easy to practice for novice dancers [12]. Its main characteristics lie in the change in dance styles according to the music, adopting rhythms, and steps from various dance styles worldwide, mainly from Latin dances [13]. In addition, Zumba is categorized as an aerobic exercise with moderate intensity [14,15]. Several previous studies have reported its validity as a preventive measure for lifestyle diseases, including reducing the body mass index (BMI), improving the levels of blood sugar and cholesterol, as well as positive effects on the cardiovascular system [16,17,18].

Mukuno et al. [9] reported that “it is rarely observed among the Philippino people to do physical or walking exercise on their own, even when they understand its benefit for their health”. Nelson [19] further indicated that the implementation rate of exercise therapy for patients with diabetes is quite low at 40–60%, and Wilson et al. [20] reported that “over 50% of individuals who take part in a fitness program will quit after the first six months.” Similarly, Nakamura et al. [21] argued that “providing an exercise program that is “enjoyable to continue” should contribute to promoting the lifestyle habit of regular health exercise.” As indicated above, most of the literature in this field has focused on the physiological effects of Zumba (e.g., BMI decrease, improvement in the blood sugar and cholesterol levels) [16,17,18]; however, there has been no investigation regarding the effects of Zumba in the Philippines, and the characteristics of the participants or factors that correlate with Zumba participation frequency remain unknown.

Moreover, there are only a few studies regarding exercise therapy in the Philippines in general; therefore, the aim of the present study was to elucidate the actual conditions of Zumba in the Philippines, along with the characteristics of its practitioners and factors related to practice participation frequency.

## 2. Materials and Methods

### 2.1. Study Design

This study was an observational and cross-sectional study.

### 2.2. Study Site and Study Period

We conducted our investigation at 10 Zumba locations (including shopping malls, sports gyms, hotels, parks, and barangays) in part of Cebu province, which included Cebu and Mactan islands (approximately 770 km from the national capital Manila, with a population of 920,000) of the Philippines for six days from 21 September to 26 September 2019. We recruited seven research collaborators, including four nutritionists from the National Nutrition Council Region VII and three local volunteers without medical licenses. The research collaborators engaged in the investigation after receiving appropriate training.

### 2.3. Inclusion Criteria and Exclusion Criteria

The inclusion criteria were being over 30 years old, and being Zumba participants living in Cebu province, Philippines. We tried to observe pre-NCDs age groups’ practice as well. The exclusion criteria were other than that. There were four people excluded, who lacked age information (one person), under 30 years old (one person), and other reasons (two persons).

### 2.4. Measurements

The data collected for this investigation included BMI derived from measurements of height (partially self-reported) and weight, along with the following items from a questionnaire survey: (1) participant characteristics (sex, age, number of family members in the same household, years of education, occupation, average monthly household income, and medical history); (2) Zumba-related details, including participation frequency, years of practice, reason(s) for starting Zumba, reason(s) for missing Zumba classes, and other related questions; (3) self-efficacy (measured by the General Self-Efficacy [22] scale); and (4) physical activities (based on the International Physical Activity Questionnaire (IPAQ) form [23]). We also added several modifications to the questionnaire items with the help of professional advice from the nutritionists recruited for this study. In the pretest, we surveyed seven Zumba participants at two locations: a shopping mall and a barangay, on 31 August 2019, with the help of a research collaborator.

We measured the participants’ height with markings made on walls or poles with a tape measure (since the use of the stadiometer was challenging). Weight was measured using two bodyweight scales: one was obtained from the principal investigator of this study, which was brought from Japan (Karada Scan 214, Omron, Kyoto, Japan), and the other was purchased locally (Electronic Body Scale Model No. EB9370, WATSONS, Hong Kong, China). We deducted 0.5 kg from the measurements taken for participants who refused to take off their shoes when being weighed. BMI was then calculated as weight (kg)/height (m)^2^, which was approximated to the second decimal place when determined using a BMI calculation software program (Keisan-CASIO, Keisan, Zaire Japan).

### 2.5. Structured Questionnaire Survey

The questionnaire survey was executed in either a one-to-one session/one-to-two session between the investigator and respondent(s) by responding to interview questions and writing responses on a survey sheet or self-written. After collecting each completed survey sheet, the investigator confirmed if there was any entry omission.

### 2.6. Data Analysis

We used SPSS software (SPSS Inc., Chicago, IL, USA) for the analyses of several variables in the univariate analysis and χ² test, including the dependent variable: “Zumba participation frequency of more/less than four times per week,” and the independent variables: age, reason(s) for starting Zumba (e.g., to enjoy dancing, to reduce weight), location (shopping mall or other), with or without fees, Zumba participation frequency of more/less than three times per week, and sex to identify correlating factors through multivariate analysis. We also confirmed the correlation with “enjoyable” as a sub-analysis through univariate analysis.

### 2.7. Ethical Considerations

This study was approved by the Ethical Review Board of Teikyo University issued on 4 September 2019 (Approval Number: 19-125). We also obtained approval in advance from the ethical committee of the Department of Health-Regional Office VII, Cebu Province, the Philippines (issued on 23 September 2019; reference number: 2019-181091051826). The field research was conducted at the various sites, and permission was obtained in advance from the management of each facility. The purpose and method of this study were orally explained to the participants. The participants were informed that this research involved an anonymous survey, which is not intended to cause any disadvantage to the participants. All participants provided written informed consent for participating in the study, and the survey was administered only to those who agreed to participate.

## 3. Results

### 3.1. Socio-Demographic Characteristics

As shown in Table 1, 197 people participated in the survey, 193 of whom completed the survey with a collection rate of 98.1%. The survey respondents consisted of 171 women (88.6%) and 22 men (11.4%), with a mean (standard deviation [SD]) age of 44.1 (8.9) years. The most common employment type was full-time work (45.1%), and the most common average monthly household income was reported among the middle class, in the range of 10,000–20,000 pesos (31.6%; approximately 200–400 USD, as of 2019). The mean years of education were 12.2, and nearly half of the participants were university graduates (43.5%). The mean number of family members living together was 4.6. The mean BMI was 25.3 kg/m^2^, and the majority of the participants (51.2%) indicated “nothing in particular” with respect to medical history, followed by hypertension (26.3%) and diabetes (5.4%). The mean self-efficacy score was 33.9 (the global average of self-efficacy = 29.55 [24]). The respondents were required to indicate the extent of physical activity that they engaged in aside from Zumba; most participants were categorized in the high physical activity group (39.9%) based on the performance of exercises, such as aerobics or jogging, aside from Zumba. The mean duration of being sedentary was 4.26 h. The results related to lesson participation frequency and months of practice showed a mean (SD) participation frequency of 3.2 (1.5) times per week and a mean Zumba participation duration of 36.1 months (median value: 24.0, interquartile range 10–60).

Table 2 presents the reasons for starting Zumba exercises. Relieving stress was the most common reason for starting Zumba (54.9%), followed by starting as a part of exercise therapy (48.2%), enjoying dancing (46.6%), and losing weight (32.1%). Being too busy was the most common reason given for being absent from Zumba class (86.3%), followed by inaccessibility to location due to the distance from residence (4.4%); no respondent indicated financial issue as the reason for absence, that is the lesson fees being too expensive.

Table 3 outlines Zumba-related questions. The majority of the respondents gave a positive answer to each question; for example, all respondents answered “yes” to the question “do you enjoy Zumba exercise?” Likewise, 49.2% of respondents answered “no” to the question “do you feel that Zumba is a hard exercise?” Additionally, 81.3% of the participants answered “yes” to the question “do you feel happy when you are practicing Zumba?” In addition, 76.6% of the participants gave a positive answer to the question, “do you want to continue Zumba in the future?” With regard to the effects of the practice, 72.3% of the participants answered that they felt positive physical effects from Zumba, and 70.7% responded positively to the question “do you feel positive mental health effects since you have started Zumba?”

### 3.2. Correlation with “Zumba Participation Frequency More than four Times per Week”

As represented in Table 4, the univariate analysis further confirmed that the variable “enjoying Zumba” was significantly correlated with the following items: positive physical effects (*p* < 0.001), positive mental health effects (*p* < 0.001), willingness to continue (*p* < 0.001), and a high level of self-efficacy (*p* < 0.001). As represented in Table 5, we investigated factors associated with the dependent variable “Zumba participation frequency more than four times per week.” The univariate analysis demonstrated that the following six items were statistically significant in the two-tailed test with a 5% significance level: older than the mean age (44.1 years) (*p* = 0.004), started Zumba to enjoy dancing (*p* < 0.001), started Zumba not to lose weight (*p* = 0.027), shopping mall location (*p* = 0.001), participation fee required (*p* = 0.017), and lesson frequency more than three times per week (*p* = 0.025). We then included these significant variables into the multivariate analysis (stepwise method) to conduct variable selection, which identified the following five variables as significant independent predictors (Table 5), older than the mean age (44.1 years) (*p* = 0.013), started Zumba to enjoy dancing (*p* = 0.001), started Zumba not to lose weight (*p* = 0.040), shopping mall location (*p* = 0.015), and participation fee required (*p* = 0.008).

## 4. Discussion

The factor “older than the mean age of all respondents” (44.1 years) showed a significant correlation with higher participation frequency (more than four times a week), suggesting that healthcare awareness tended to increase for individuals in their 40s, which is in line with existing survey data on the Philippines [25]. The participants in that age group are more likely to have completed raising young children and therefore have more time to spare. With respect to the questions regarding Zumba (Table 4), all respondents answered that Zumba was enjoyable, and the following reasons for practicing Zumba: “to enjoy dancing” but not “to lose weight,” were significantly correlated with participation frequency, which was similar to previous studies that showed that the “enjoyable” factor correlated with the habit of regular exercise [21,26]. Although the overall response rate was low, other reason(s) for starting Zumba such as “to make more friends” (12.4%), “to enjoy music” (11.4%), or “to communicate with more people” (5.7%) were likely correlated with the enjoyment factor, supporting similar results from a previous study [27] that citizen participation in dance-type exercises shows higher levels of community involvement and enjoyment than that found in other types of exercise therapies (Table 2).

More than 80% of the respondents answered that they did not consider Zumba as a hard exercise (Table 3). The fact that Zumba is categorized as moderate-intensity exercise therapy [14,15] should motivate continuation, as shown in a previous study that physical competence, “capability”, should correlate with the continuation of Zumba [21].

The realization of physical effects and mental health effects, willingness to continue and a high level of self-efficacy showed significant correlations with the enjoyment factor for Zumba (Table 4), supporting previous studies showing that dancing should bring about positive influences on people’s physical and mental status [28,29] and that a high level of self-efficacy should influence people’s desire for exercise continuation and practice [30,31,32].

Regarding the lesson location (Table 5), the results suggested that increasing the lesson frequency at each location should contribute to increasing participation frequency, given its significant correlation with a lesson frequency of more than three times per week. In addition, lessons that took place in a shopping mall location were significantly correlated with higher participation frequency, indicating the relevance of location accessibility. Shopping mall facilities are relatively common in the Philippines, serving as a popular place for meeting and exchange in the neighborhood. In contrast to visiting a specific location for Zumba, participants can drop by a shopping mall Zumba location in the course of their shopping. In addition, since people of various strata visit shopping malls, having a Zumba facility in this location would enhance Zumba exposure, probably serving as a key factor for a population approach to involve a far larger population in practice. The requirement of the participation fee (10–20 pesos) also showed a significant correlation with participation frequency. This fee is quite low, equivalent to the price of a piece of bread or a can of soft drink; paying the participation fee should be correlated with the participants’ motivation or eagerness to practice. However, we cannot entirely deny the possibility of reverse causation in this regard; therefore, further investigation is necessary. Nabetani et al. [33] argued that “for most of us, it is more challenging to continue exercises than to start it,” and according to Wilson et al. [20], “over 50% of individuals who take part in a fitness program will drop out after the first six months.” Both of these studies highlight how hard it is to continue exercises.

Respondents in this study gave positive answers to the survey questions indicating that the Zumba was “very enjoyable” or “enjoyable”, including those with medical histories of hypertension or diabetes who were participating as a part of exercise therapy, or those with high BMI levels who participated in reducing weight. All participants also realized the positive physiological/psychological effects of Zumba and indicated their eagerness to continue the exercise. In addition, the mean participation frequency and months of continued practice both showed a tendency to be high among the majority of participants, suggesting that their exercise activity developed into a habit. The participation of groups of various ages (up to those in their 80 s), income levels, or academic backgrounds demonstrates that Zumba implementation in the Philippines is a successful exercise therapy that is enjoyable for the participants, as well as a successful population approach that can be widely accepted and practiced regularly among people, with additional advantages that it is provided for free or at quite inexpensive fees at various public places, including shopping malls, parks, hotels, and barangays.

The main limitations of our study were as follows. First, we cannot generalize the results because this was a cross-sectional study conducted only on Zumba locations in the Cebu province of the Philippines. Further, the number of male participants was too small. Second, there was potential for selection bias due to only including individuals already practicing Zumba, along with those who had already been practicing for a long time and at a high frequency. Third, the causal relationship among the factors remains unknown since this was a cross-sectional investigation. Despite these limitations, few field studies have been implemented on a population approach for lifestyle disease prevention measures in the Philippines, and our study, therefore, has the advantage of being the first field study in the world implemented in a real-life situation.

## 5. Conclusions

This study revealed that the element of “enjoyable”, as well as good locations (accessible) at various public places and low participation fees, might strongly influence the continuation and frequent participation of Zumba exercise, which is a successful population approach that can be widely accepted and practiced regularly among people in the Philippines. These initiatives may induce positive inhibition of the increase in NCD, such as diabetes in this country.

## Figures and Tables

**Table 1 ijerph-18-05339-t001:** Characteristics of Zumba participants (*n* = 193).

Variable		*N*	%
**Sex**	Female	171	88.6
	Male	22	11.4
**Age (years)**	30–39	65	33.7
Mean: 44.1(8.9)	40–49	70	36.3
Minimum Value: 30	50–59	50	25.9
Maximum Value: 81	60–69	6	3.1
	≥70	2	1.0
**Occupation** [*n* = 191]	Full-time work (Office work)	87 (32)	45.1 (16.6)
	Part-time work	19	9.8
	Housewife	39	20.2
	Other work	20	10.4
	None	26	13.5
**Monthly Family Income** [*n* = 190]	None	6	3.1
(Philippines Peso)	0 < 10,000	42	21.8
(1 peso = about 0.02 US$)	10,000–20,000	61	31.6
	20,000–40,000	60	31.1
	>40,000	21	10.9
**Final Education** [*n* = 192]	University completed	84	43.5
Average of schooling years:	University uncompleted	47	24.4
12.2 (2.0)	High school completed	44	22.8
	High school uncompleted	15	7.8
	Primary school completed	2	1.0
**Number of family members**	1	11	5.7
(Total individuals) [*n* = 185]	2–4	95	49.2
Mean: 4.6	5–9	71	36.8
	10≥	8	4.1
**Medical History**	Diabetes	11	5.4
	Hypertension	54	26.3
	Heart Diseases	7	3.4
	Cancer	1	0.5
	Gastric Problem	5	2.4
	Other Diseases	22	10.7
	None	105	51.2
^1^ **BMI** [*n* = 189]	17.0–18.5	6	3.2
Mean: 25.3 (4.13)	18.5–25.0	88	46.6
Minimum Value: 17.0	25.0–30.0	70	37.0
Maximum Value: 38.3	30.0–35.0	21	11.1
	35.0–40.0	4	2.1
^2^ **GSE** [*n* = 188]	≤19	1	0.5
[Score: 10–40]	20–24	4	2.1
Mean: 33.9 (4.1)	25–29	17	9.0
Minimum Value: 13	30–34	71	37.8
Maximum Value: 40	35–40	95	50.5
^3^ **IPAQ** [*n* = 173]	Low physical activity	56	32.4
Excluding Zumba activity	Moderatephysical activity	48	27.7
	Vigorousphysical activity	69	39.9
**Sitting Time** (Hour) [*n* = 183]	<3	65	39.1
Mean: 4.26	3–5	54	32.4
	6–10	48	24.0
	10≥	16	9.6
**Frequency of Zumba participation** [*n* = 192]	<1/week	9	4.7
Mean: 3.2 times/week (SD:1.5)	1/week	18	9.4
	2/week	46	24.0
	3/week	62	32.3
	4/week	22	11.5
	5/week	20	10.4
	6/week	5	2.6
	7/week	10	5.2
**Duration of** **Zumba participation** [*n* = 193]	<6 months	33	17.1
	6 months–1 year	27	14.0
	1–5 years	82	42.5
	5–10 years	41	21.2
	≥10 years	10	5.2

^1^ BMI: Body Mass Index, ^2^ GSE: General Self-Efficacy, ^3^ IPAQ: International Physical Activity Questionnaire.

**Table 2 ijerph-18-05339-t002:** Reasons for starting Zumba classes (*n* = 193) and for absence from Zumba classes (*n* = 183).

Reason for Starting Zumba Classes (Multiple Answers)	Number of Responses (%)	Reason for Absence from Zumba Class	*n* (%)
Relieve stress	106 (54.9)	Busy	158 (86.3)
For exercise therapy	93 (48.2)	Far from your place	8 (4.4)
Enjoy dancing	90 (46.6)	Attendance fee is high	0 (0.0)
Lose weight	62 (32.1)	Other reasons	17 (9.3)
Make more friends	24 (12.4)		
Enjoy Music	22 (11.4)		
Communicate with more people	11 (5.7)		
Other reasons	4 (2.1)		
Respect instructor	2 (1.0)		

**Table 3 ijerph-18-05339-t003:** Feeling about Zumba class (*n* = 193).

Feeling about Zumba Class	Strongly Agree	Agree	Disagree	Strongly Disagree
**Do you enjoy Zumba exercise?**	151 (78.2)	42 (21.8)	0 (0.0)	0 (0.0)
**Do you think Zumba is a hard exercise?**	14 (7.3)	19 (9.8)	95 (49.2)	65 (33.7)
**Do you feel happy when you are doing Zumba?**	157 (81.3)	35 (18.1)	0 (0.0)	1 (0.5)
**Do you want to continue Zumba in the future?**	147 (76.6)	43 (22.4)	2 (1.0)	0 (0.0)
**Did you feel the positive physical effects since you have started Zumba?**	138 (72.3)	50 (26.2)	3 (1.6)	0 (0.0)
**Did you feel the positive mental health effects since you have started Zumba?**	135 (70.7)	54 (28.3)	2 (1.0)	0 (0.0)

**Table 4 ijerph-18-05339-t004:** Factors associated with the Question: “Do you enjoy Zumba exercise?”: Results of univariate analysis.

Variable		Very Enjoyable (*N* = 151)		Enjoyable (*N* = 42)		OR	95% CI	*p*-Value
		**n**	**%**	**n**	**%**			
**Physical effects**	Very effective	124	83.8	14	35.0	9.60	4.39–21.00	<0.001
	Effective	24	16.2	26	65.0			
**Mental health effects**	Very effective	123	82.0	12	30.8	10.25	4.62–22.75	<0.001
	Effective	27	18.0	27	69.2			
**Willingness to continue**	Strongly agree	130	86.1	17	43.6	8.01	3.66–17.53	<0.001
	Agree	21	13.9	22	56.4			
**GSE**	≥34 ^a^	99	65.6	14	33.3	3.81	1.85–7.85	<0.001
	<34	52	34.4	28	66.7			

^a^: Average GSE of respondents: 34; OR, odds ratio; CI, confidence interval; GSE, general self-efficacy.

**Table 5 ijerph-18-05339-t005:** Factors associated with the frequency of participation in Zumba exercise (≥4 times/week): Results of logistic reguression analysis.

		≥4 Times/Week		<4 Times/Week		Univariate Analysis				Multivariate Analysis			
		(*N* = 58)		(*N* = 135)									
		*n*	%	*n*	%	COR	95% CI		*p*-Value	AOR	95% CI		*p*-Value
Age, yrs	≥ 44 ^a^	35	60.3	51	37.8	2.51	1.33	4.71	0.004	2.47	1.21	5.05	0.013
	<44	23	39.7	84	62.2	1.00				1.00			
Sex	Female	51	87.9	120	88.9	0.91	0.35	2.37	0.848				
	Male	7	12.1	15	11.1	1.00							
Reason for starting Zumba (1)													
Enjoy Dancing		39	67.2	45	35.4	3.74	1.94	7.22	<0.001	3.68	1.81	7.48	<0.001
None of the above reasons		19	32.8	82	64.6	1.00				1.00			
Reason for starting Zumba (2)													
Lose weight		12	20.7	47	37.0	0.44	0.21	0.92	0.027	0.42	0.19	0.96	0.040
None of the above reasons		46	79.3	80	63.0	1.00				1.00			
Location													
Shopping Mall		37	63.8	52	38.5	2.81	1.49	5.32	0.001	2.41	1.18	4.91	0.015
Other than the shopping mall		21	36.2	83	61.5	1.00				1.00			
Participation Fee													
Paid		28	48.3	41	30.4	2.14	1.14	4.03	0.017	2.60	1.28	5.29	0.008
Free		30	51.7	94	69.6	1.00				1.00			
Number of Times Held													
≥3 Times/week		41	70.7	72	53.3	2.11	1.09	4.08	0.025				
<3 Times/week		17	29.3	63	46.7	1.00							

^a^: average age of participants: 44.1 years; Multivariate analysis (Stepwise Method): Adjusted for Age, Sex, Reason for starting Zumba (enjoy dancing, lose weight); Held in shopping mall, Participation Fee, and Number of Times held; COR, crude odds ratio; AOR, adjusted odds ratio; CI, confidence interval.

## Data Availability

The data are not publicly available due to privacy/ethical restrictions.

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
