# Peer review of "Factors Associated with Regular Zumba Practice as Preliminary Results: A Population-Based Approach in Cebu Province, the Philippines"

_ijerph, 2021, doi:10.3390/ijerph18105339_

Round 1

Reviewer 1 Report

The manuscript is well written and very interesting.

Material and methods:

A lack of information about the voluntary participation in the study.

Why did the authors decide to interview  one-to-one and was the questionnaire filled out by the reviewer? Sometimes the participants do not feel anonymous is such a situation. Please explain.

What was the inclusion and exclusion criteria? How many participants were excluded and why?

How was the IPAQ assessed?

Discussion:

I do not recommend dividing the discussion into sub-divisions and referencing it to the tables. This is typical for results.  

Conclusions:

Conclusions are clear, however they  need to be more descriptive and respond more to the purpose of the study.

Reviewer 2 Report

According to the justified explanation of the Authors (line 264-273),  “First, we cannot generalize the  results because this was a cross-sectional study conducted only on Zumba locations in the Cebu province of the Philippines. Further, the number of male participants was too small.  Second, there was potential for selection bias due to only including individuals already  practicing Zumba, along with those who had already been practicing for a long time and at a high frequency. Third, the causal relationship among the factors remains unknown since this was a cross-sectional investigation. Despite these limitations, few field studies  have been implemented on a population approach for lifestyle disease prevention  measures in the Philippines, and our study therefore has the advantage of being the first  field study in the world implemented in a real-life situation”,

The title needs to be changed:

  • should contain the phrase "preliminary results".
  • the statement "in Cebu and Mactan” could be changed on  “Cebu province."
  • the statement "to reduce prevalence of non-communicable diseases" should be removed, because the impact of Zumba practice in reducing the risk of these diseases has not been investigated.

Materials and Methods

 Page 2, line 87 - the phrase “In Cebu province” needs to be completed, for example “In part of Cebu province, which included Cebu and Mactan  islands”

Page 6, line 181 -  It should be  Table 5  instead  of Table 4

 Line 192  - information  “Table 4” should be added.

Conclusion

The conclusions can be supplemented with other important factors related to the regular Zumba practice, as mentioned in chapter Results and appropriate Tables. 
